# Psychological Intervention for Patients with Biopsychosocial Late Effects Following Surgery for Colorectal Cancer with Peritoneal Metastases—A Feasibility Study

**DOI:** 10.3390/cancers17071127

**Published:** 2025-03-27

**Authors:** Rogini Balachandran, Henriette Vind Thaysen, Peter Christensen, Eva Rames Nissen, Mia Skytte O’Toole, Sofie Møgelberg Knutzen, Cecilie Dorthea Rask Buskbjerg, Lisa Maria Wu, Nina Tauber, Ali Amidi, Josefine Tingdal Taube Danielsen, Robert Zachariae, Lene Hjerrild Iversen

**Affiliations:** 1Department of Surgery, Aarhus University Hospital, 8200 Aarhus, Denmark; 2Department of Clinical Medicine, Aarhus University, 8000 Aarhus, Denmark; 3Danish Cancer Society Centre for Research on Survivorship and Late Adverse Effects After Cancer in the Pelvic Organs, 8200 Aarhus, Denmark; 4Unit for Psychooncology and Health Psychology, Aarhus University Hospital, 8200 Aarhus, Denmark; 5Department of Oncology, Aarhus University Hospital, 8200 Aarhus, Denmark; 6Department of Psychology and Behavioral Sciences, Aarhus University, 8000 Aarhus, Denmark; 7Department of Psychology, Reykjavik University, 101 Reykjavik, Iceland; 8Danish Center for Breast Cancer Late Effects (DCCL), 8200 Aarhus, Denmark

**Keywords:** colorectal cancer, peritoneal metastases, late effects, biopsychosocial, psychological intervention

## Abstract

Up to 80% of patients experience late effects (LEs) one year after surgery for peritoneal metastases from colorectal cancer. In this study, we tested the feasibility and outcome of a treatment strategy to address LEs. Patients from Denmark were screened for biopsychosocial LEs (anxiety, depression, fear of cancer recurrence, insomnia, cognitive impairment, pain, fatigue). Patients scoring according to clinical cut-offs were referred to a Multi-Disciplinary Team (MDT) conference, in which a personalized intervention was decided on. The patients participated in this conference and later on received the decided intervention. We found a significant improvement in their “Measure Yourself Concerns and Wellbeing” scores following the intervention. We conclude that screening for LEs and conducting an MDT conference can provide a personalized intervention plan, which patients are able to complete and may benefit from. Testing this approach in a larger patient population and for different tumor origins is important.

## 1. Introduction

It is well-known that cancer survivors often develop biopsychosocial late effects (LEs) such as anxiety, depression, cognitive impairment, fear of cancer recurrence (FCR), pain, fatigue, and insomnia [1,2,3,4,5,6,7,8,9]. For colorectal cancer (CRC) patients (stage I-III), the literature reports that many of these patients may suffer from poor sleep quality, which is associated with an increased incidence of anxiety and depression [3]. FCR is another well-known LE in CRC survivors, with a review from 2021 reporting that up to 16% of patients experience a high level of FCR [5]. Cognitive impairment, e.g., difficulties with concentration and memory, is another LE particularly affecting those returning to the labor market [8]. In a recent paper, we investigated these and other LEs in CRC stage IV patients [10]. We found that up to 80% of patients who underwent curatively intended surgery for CRC with peritoneal metastases (PM) reported biopsychosocial LEs one year after the surgery. In addition, patients who developed moderate to severe LEs experienced a poorer quality of life (QoL) than both patients who developed no to mild LEs and the general Danish norm population [11]. Thus, screening for and treating biopsychosocial LEs ought to be included as part of regular, comprehensive follow-up care for cancer patients.

The use of patient-reported outcome measures (PROMs) offers clinicians a convenient and rigorous way to screen and track LEs over time [12]. Indeed, recent guidelines published by the European Society for Medical Oncology (ESMO) recommended LE monitoring in cancer survivors [13,14], especially in patients with a high risk of recurrence and/or treatment-related side-effects [14]. Patients with PM from CRC (CRC-PM) belong to such a group and are therefore prime candidates for such monitoring. Furthermore, a recent review that focused on the management of LEs recommended that biopsychosocial LEs be managed by cognitive and behavioral interventions, preferably by psychologists experienced in cancer survivorship [15].

Surgery for CRC-PM is often centralized due to the complexity of the surgery. Thus, many patients may live far away from treatment centers making it difficult to deliver LE interventions in a face-to-face setting. It is therefore highly relevant to also investigate the delivery of psychological online interventions. Different modes of delivery, e.g., group format [16,17], online delivery [18,19], or delivery via telephone [20] have been investigated before and proven to be just as effective or even better than face-to-face delivery. However, these studies focused on other cancer populations and on patients with less aggressive disease.

The present study aimed to assess the feasibility of (1) a clinical assessment procedure to screen cancer survivors for selected biopsychosocial LEs (anxiety, depression, cognitive impairment, FCR, pain, fatigue, and insomnia) following surgery for CRC-PM, (2) patient participation in an online multidisciplinary team (MDT) conference, and (3) online delivery of a personalized behavioral intervention targeted at the most debilitating LEs. Furthermore, we wanted to evaluate the preliminary efficacy of the intervention and to describe patients’ experiences of the MDT conference and the intervention.

## 2. Materials and Methods

### 2.1. Study Design, Population, and Setting

The Department of Surgery, Aarhus University Hospital, is the only center performing intended curative surgery, namely cytoreductive surgery with hyperthermic intraperitoneal chemotherapy, for CRC-PM in Denmark. The PROMs-based screening procedure was undertaken between January 2021 and May 2023 in order to determine the nature and severity of selected LEs, and whether they exceeded clinical cutoff scores. The selected LEs were chosen based on the existing literature as to which LEs may potentially exist in cancer survivors. Two screening approaches were undertaken: (1) prospective screening in which PROMs were administered to newly operated patients 3, 6, and 12 months after surgery; and (2) cross-sectional screening in which patients who were at least one year post-surgery were administered the PROMs once during a follow-up appointment (18, 24, 36, 48 or 60 months after surgery), see Figure 1. PROMs were administered pre-intervention, one month post-intervention, and six months post-intervention.

Patients who exceeded established clinical cutoff scores were offered a referral to the MDT conference, and subsequent psychological intervention if deemed relevant by both patients and clinicians (see Figure 2). Patients were excluded from the study if they (1) were patients currently receiving psychological services, (2) experienced psychological and physical symptoms exclusively unrelated to their cancer treatment, and (3) experienced recurrence of their disease within a year following their surgery, since a substantial proportion of patients develop a recurrence within the first year after surgery [21,22].

### 2.2. PROMs, Cutoff-Levels, and Intervention

The inclusion criteria were age ≥ 18 years, the ability to understand written Danish and undergoing curatively intended surgery for CRC stage IV (peritoneal metastases). The exclusion criteria were patients undergoing surgery for cancers other than CRC stage IV. See Table 1 for which (1) LEs were investigated [23,24,25,26,27,28,29,30,31], (2) the PROMs used for screening, (3) referral eligibility to the MDT conference and (4) the specific intervention components. All patients were also administered the ‘Measure Yourself Concerns and Wellbeing” (MYCaW^®^) questionnaire [32,33]. This is a PROM designed for evaluating holistic and personalized approaches to supporting people. MYCaW focuses on the concerns that are of highest priority to the individual, as defined by the patient. The patient selects the two most debilitating LEs (primary LE and secondary LE) and rates them and their general wellbeing on a 7-point Likert scale (0–6). For the primary and secondary LEs, larger scores indicate greater severity. For general wellbeing, larger scores indicate poorer wellbeing. MYCaW has been translated to and validated in Danish [34].

### 2.3. The MDT Conference

During each monthly MDT conference, we discussed with one or two patients, with one hour allocated to each patient. The conference was held online using ‘Rooms’, a secure platform for video conferences, distributed by the Central Region in Denmark. Prior to the conference, the principal investigator (RB) guided the patient in how to navigate and log into the platform. Participants included the patient, RB, colorectal surgeon(s), nurse specialist(s), and a group of clinical psychologists with expertise in psycho-oncology. The RB briefly presented the patient’s history and reasons for referral. The patient then described their primary and secondary LEs. Then, one of the psychologists conducted an interview to confirm the primary and secondary LEs which were the patient’s most debilitating issue(s) and to identify a key symptom (e.g., low mood), and then hypothesized maintaining cognitive and behavioral processes (e.g., inactivity). Next, with the patient still present, the attending psychologists discussed the patient’s symptoms and proposed a personalized “treatment package” of cognitive and/or behavioral intervention components. Finally, the patient was offered a personalized intervention.

### 2.4. Psychological Intervention

The intervention was delivered by one of the psychologists with expertise in cancer survivorship and the specific LEs investigated in the present study. The intervention offered was a combination of cognitive and/or behavioral strategies previously proven efficacious for the identified LEs [35,36,37,38,39,40,41,42,43,44,45,46,47,48,49], (see Table 1). Up to six 1 h sessions were offered online. The aim of the intervention was to treat the primary and secondary LEs as identified by the MYCaW and teach the patient new strategies for coping with these symptoms through in-session training and homework assignments. If patients developed a recurrence of their cancer during the intervention, they were excluded from the study but offered continued counseling up to a maximum of 6 total sessions with the psychologist.

### 2.5. Patient Satisfaction

Once the patient had completed the post-intervention PROMs, the RB contacted the patient and conducted a semi-structured interview by telephone. After intervention, semi-structured interviews were conducted with the main purpose of consecutively exploring areas that could be improved regarding the conference and/or the sessions. The interview guide contained questions pertaining to the overall experience of participating in the MDT, the sessions, whether the sessions were experienced as effective, and if this approach was something that they would recommend to other patients. Further, questions were asked about which aspects worked well, and which did not work well. The first three patient interviews were analyzed by the RB and one more colleague (HVT) with experience in semi-structured interviews. The remaining interviews were only evaluated by the RB, who in case of doubt corresponded with the HVT.

### 2.6. Data Analysis

The descriptive statistics included the proportion of (1) eligible patients who (2) accepted referral, (3) participated in the MDT conference, (4) initiated the intervention, and (5) completed the intervention. The total score of each outcome was evaluated for all participating patients pre- and post-intervention. The mean pre–post change scores of the LE outcomes and MYCaW measure, as well as the associated 95% confidence intervals, were calculated, and differences in scores from pre- to post-intervention were analyzed with parametric (paired *t*-tests) or non-parametric tests (Wilcoxon signed-rank tests), as appropriate. A *p*-value ≤ 0.05 was significant. Further, the effect size was estimated with Hedges’s g where a positive value indicated a beneficial effect of the intervention. A value of 0.2 indicated a small effect, a value of 0.5 indicated a medium effect, and values ≥ 0.8 indicated a large effect.

### 2.7. Ethical Approvals

The study was registered with the Danish Data Protection Agency (#1-16-02-714-20) and preregistered with ClinicalTrials.gov (#NCT04956107). The local scientific ethics committee of the Central Region in Denmark concluded that the study was exempt from formal approval by a scientific ethics committee (#1-10-72-11-22). All patients provided oral and written consent before inclusion in the study.

## 3. Results

### 3.1. Feasibility and Acceptability

From January 2021 to May 2023, a total of 117 patients were screened. Twenty-eight patients (24%) were eligible for referral to the MDT conference, and 13 (46%) were referred, see Figure 3. The mean age for referred patients was 59 years (range 39–72) and 85% were females. All referred patients participated in an MDT conference conducted from June 2022 to May 2023 and were offered a personalized online intervention. By the end of July 2023, eleven patients had completed the offered intervention. All patients requested the intervention in an online format. One patient developed a recurrence shortly after the last session and did not complete the post-intervention PROMs. MDT conferences lasted an average of 47 min (range 38–58) and between two and six psychologists participated in each MDT. According to MYCaW, the primary LEs were fatigue (*n* = 4), cognitive impairment (*n* = 3), FCR (*n* = 3), pain (*n* = 1), and insomnia (*n* = 1). The secondary LEs were fatigue (*n* = 4), cognitive impairment (*n* = 2), insomnia (*n* = 2), anxiety (*n* = 2), depression (*n* = 1), and pain (*n* = 1).

Patients who declined participation were, in general, older (mean age 66 (range 53–79) and primarily men). (73%). See Appendix A for differences between participants and non-participants.

### 3.2. Outcome of the Intervention

Statistically significant pre- to post-intervention improvements were observed for all three items in the MYCaW both 1 month and 6 months after the intervention. Effect sizes were similar (>0.8) for all three MYCaW items both 1 and 6 months after the intervention, indicating a large effect of the intervention (see Table 2). 

One month post-intervention, nine out of ten participants reported improvements in their primary LE and one patient reported no change. Likewise, for the secondary LE, nine out of ten reported an improvement while one patient did not. Seven out of ten reported improved general wellbeing, and three out of ten had a stable score. In total, six patients completed the 6 months post-intervention PROMs. At that timepoint, five out of six reported improvements in their primary LE and one patient reported no change. For the secondary LE, five out of six reported an improvement and one patient again had a stable score, and for general wellbeing four out of six reported an improvement and two patients had a stable score. No patient reported worsened MYCaW scores from pre- to 1 month post-intervention or from pre- to 6 months post-intervention.

Anxiety and insomnia scores improved significantly from pre- to 1 month post-intervention. One month after the intervention, reduced symptom scores were also observed for remaining LEs such as depression, FCR, fatigue, pain, and cognitive impairment, but did not reach statistical significance. Six months post-intervention, reduced symptom scores were observed for all LEs except for fatigue. Again, these improvements did not reach statistical significance (see Table 2).

### 3.3. Patient Satisfaction

The semi-structured interviews were conducted with the main purpose of consecutively exploring areas that could be improved in regard to the conference and/or the sessions. The interviews revealed three themes that pertained to the: (1) overall experience of the MDT conference, (2) overall experience of the sessions, and the (3) overall evaluation and conclusion. Patients expressed satisfaction with the MDT conference and the intervention. All patients, except one, expressed satisfaction with the online format of both the MDT conference and the sessions. Most patients also mentioned that the possibility of Appendix A for further details).

## 4. Discussion

We aimed to assess the feasibility, acceptability, and preliminary efficacy of participating in an online MDT conference and psychological intervention addressing the primary and secondary LEs which patients wanted help with following intended curative surgery for CRC-PM. Of the patients who were eligible for referral, almost half (46%) were referred. Eligibility was based on predefined cut-off values. A stricter cut-off would potentially risk excluding some patients in need, and a more lenient cut-off would potentially increase the number of eligible patients not being impacted to a sufficient extent. Patient acceptance of referral was obtained via a phone call. Overall, the burden of making phone calls to patients was minimal in terms of time and cost, and this is why we argue that the selected cut-offs were appropriate.

Screening for LEs and treating LEs should be an important part of the follow-up treatment offered to cancer survivors. PROMs offer a systematic and timely approach to achieve this [12]. The systematic approach towards screening and treating LEs used in the present study ensured that the risk of having undetected LEs was low. Referral based solely on patients’ reports of complaints may not have achieved the same level of comprehensive care. The PROMs selected for this study were selected due to their precision and comprehensiveness, compared to a more general PROM with only one–two questions for each LE investigated. Further, all the chosen PROMs had defined cut-off levels and hence could be used for monitoring the effect of the intervention.

The fact that approximately half of the eligible patients declined referral to the MDT emphasizes that having an LE based on a score is not the same as having an LE which affects you to a degree where help is needed or wanted. The main reason why patients declined referral to the MDT was due to “not being significantly impacted” by the LEs, which aligns with the existing literature showing this as a common explanation for non-participation in psychological interventions for distress among cancer survivors [50,51]. Non-participants were predominantly men, a finding consistent with previous studies of patient participation in psychological interventions [50,51,52]. Among patients who accepted referral, all patients participated in the MDT, and all patients except one initiated the intervention. The acceptance rate of the intervention surpassed the rates in the existing literature, where less than half of distressed patients (29–43%) expressed interest in receiving help following cancer treatment [4,45]. Notably, practical factors such as physical or geographical constraints commonly hinder participation [50,51]. Thus, it is possible that the online option facilitated participation amongst our participants. The disadvantages of having an online setup could include difficulties with managing the online platform for some patients. However, all patients in this study managed the online setup with the help of the PI. Another possible disadvantage of the online setup could be the lack of physical interaction between the patient and psychologist. However, previous studies have shown that online and phone-based interventions are just as effective as face-to-face interventions [18,19,20].

The literature identifies additional reasons for non-participation in psychological interventions, such as patients’ preference for the self-management of their problems [51,53] and concerns regarding the stigma associated with seeking psychological treatment [53,54]. In our study, conducting an MDT conference prior to the intervention, with the participation of the patient’s surgeon or nurse specialist, aimed to alleviate stigma by familiarizing patients with the psychological process and provided an environment of safety and alignment of expectations. The participants were made aware of the focus of the sessions. This geared them towards being equipped with tools to help them improve their symptoms. The MDT, therefore, served as a form of introduction to the therapeutic process, and participants provided positive feedback regarding their experience of the MDT in the semi-structured patient interviews.

Beyond investigating feasibility, we also investigated the preliminary efficacy of the offered interventions, and the results demonstrated statistically significant patient reported improvements in general wellbeing and the two most debilitating (primary and secondary) LEs experienced by the patients. The interventions offered are known to be efficacious [35,36,37,38,39,40,41,42,43,44,45,46,47,48,49], however their efficacy has never been tested in an online setup for patients with such advanced disease as CRC-PM. Another noticeable factor regarding the preliminary efficacy is that the significant improvement seen for all three items in MYCaW was not demonstrated for the symptom-specific PROMs. The lack of significant results for the symptom-specific PROMs could be due to the low number of patients (type 2 error). However, it raises the question of whether improvement in the symptom-specific PROMs or improvement in MYCaW scores, in which patients report that they experience an improvement, matters the most. The authors of this paper would argue that the patients’ subjective experience of improvement is the most important factor. Experiencing an improvement in the symptom-specific PROMs without the patients reporting an improvement does not seem to be beneficial for the patient. Hence, including a PROM based on self-evaluated improvement is important.

### 4.1. Sustainability and Scalability

When looking at the sustainability and scalability of the approach described in this study, several aspects have to be considered. First, the costs of this approach should be considered. As mentioned before, the conferences and subsequent intervention were not without cost. However, as the psychologists participating in this study each had expertise regarding several LEs, it was sometimes feasible to conduct the conference with only two participating psychologists. For the patient, costs were negligible, as both the MDT conference and the intervention were offered in an online format free of charge. With increasing experience, the MDT conferences could, most likely, be conducted faster. For some of the LEs, the intervention could be offered in an online group format, which has shown comparable effectiveness [17,39]. Conducting MDT conferences also provided the psychologists with the opportunity to assess the patient before initiating the intervention, thereby saving time and facilitating identification of strategies expected to benefit the patient the most. Identifying this prior to the intervention has been shown to increase the proportion of patients initiating such interventions [55]. Second, the approach described in this study should be reserved for patients with moderate–severe LEs. Mild LEs could possibly be managed by self-help tools or by psychologists or doctors with less experience in psycho-oncology. Reserving this approach for patients with moderate–severe LEs ensures that only patients with the most debilitating LEs are offered intervention with a psychologist with expertise in psycho-oncology. Third, another factor contributing to the scalability of this study is that the investigated LEs are generic and presumed to exist across different cancer types. Hence, the results of this study could be implementable in other patient populations. Fourth, exploring the reasons behind patients declining participation in an MDT conference and addressing their unmet needs would be highly relevant.

The intervention in the present study (conducting an online MDT conference and online personalized psychological intervention addressing the most debilitating LEs) is part of an ongoing Danish national study for CRC stage I–II. The study is called RESPONSE: Colorectal Cancer Survivors’ Follow-up Care—Now Digital and Need-based. The overall aim of this study is to investigate whether a change to the current follow-up program following treatment for CRC stage I–II could improve health-related QoL without compromising overall survival, recurrence-free survival or increasing costs. As part of the change in the current follow-up program, the approach described in the present feasibility study will be further evaluated. This study has been preregistered with ClinicalTrials.gov (trial #: NCT06614647).

### 4.2. Study Limitations

The limitations of our study should be noted. First, given the nature of the study design, including the small sample size, the results ought to be considered preliminary. Second, long-term follow-up (beyond six months) was not conducted on these patients, highlighting the need to investigate the long-term effectiveness of the offered intervention strategy. Third, although it is an important aspect of feasibility, cost-effectiveness was not investigated. Conducting MDT conferences with several participating clinical psychologists and carrying out the subsequent interventions can be costly. Fourth, we did not explore whether patients declining participation in an MDT conference differed in other factors besides age and gender. In the literature, participants have been described as more distressed, more likely to be employed, younger, higher educated, or more recently diagnosed with their cancer diagnosis [54]. Lastly, we did not evaluate treatment fidelity, meaning we did not systematically assess whether the intervention strategies were implemented as intended.

## 5. Conclusions

The systematic approach tested in the present study for screening and treating various biopsychosocial LEs was shown to be feasible, acceptable, and demonstrated a positive preliminary efficacy. Patients successfully engaged in the MDT conferences, initiated, and completed the personalized interventions offered. The next step is to investigate the systematic approach outlined in the present study across larger patient populations encompassing diverse cancer types.

## Figures and Tables

**Figure 1 cancers-17-01127-f001:**
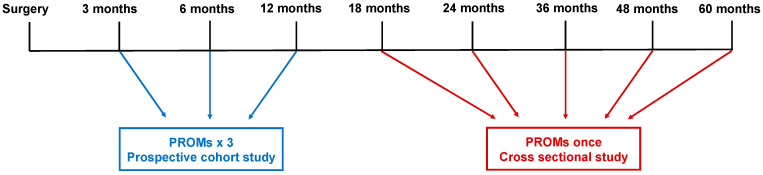
Timing of PROMs.

**Figure 2 cancers-17-01127-f002:**
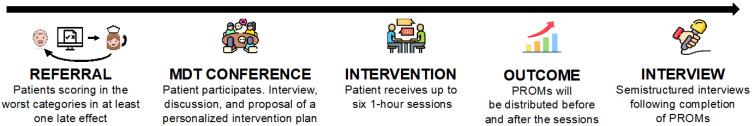
Time and specific components of the present study.

**Figure 3 cancers-17-01127-f003:**
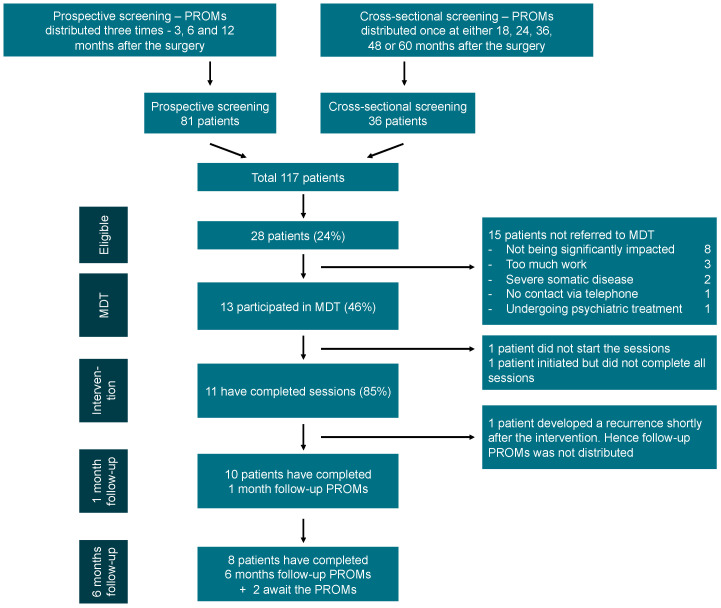
Patient flow diagram.

**Table 1 cancers-17-01127-t001:** Referral eligibility and interventions.

Late Effect	Questionnaire and Eligibility for Referral	Intervention
Anxiety	Generalized Anxiety Disorder-7 (GAD-7) [22]. Cut-off ≥ 10: moderate to severe anxiety	Cognitive behavioral therapy (CBT) and mindfulness-based intervention strategies, including value-based activation, decentering, and attention training [34,35,36].
Depression	Patient Health Questionnaire-p (PHQ-9) [23]. Cut-off ≥ 10: moderate to severe depression	Cognitive behavioral therapy (CBT) and mindfulness-based intervention strategies, including value-based activation, decentering, and attention training [34,35,36].
Fear of Cancer Recurrence (FCR)	FCR Inventory—short form (FCRI-SF) [25,26] Cut-off ≥ 22: severe FCR	Meta-cognitive therapy including strategies for reducing worry and excessive threat monitoring, modifying unhelpful meta-cognitive beliefs, and promoting value-based living [37,38,39,40].
Insomnia	The Insomnia severity index (ISI) [27]. Cut-off ≥ 15: moderate to severe insomnia	Cognitive behavioral therapy for insomnia (CBT-I) components include: sleep restriction, relaxation training, stimulus-control therapy, cognitive therapy targeting maladaptive cognitions about sleep, rumination, and worrying, and sleep hygiene education [41,42].
Fatigue	Functional Assessment of Chronic Illness Therapy—Fatigue’ (FACIT-F) [28,29]. Cut-off ≤33: moderate or severe fatigue	Treating other factors that may contribute to fatigue, e.g., insomnia. If fatigue persisted, once other causes had been eliminated, other approaches for managing fatigue were implemented, including exercise, energy management, bright white light (BWL) therapy [43,44,45].
Pain	The Rectal cancer pain score [30]. Cut-off ≥ 18: pain with major impact on quality of life	Mindfulness-Based Cognitive Therapy and mind–body therapy-based strategies for reducing pain through relaxation training and reducing pain catastrophizing through attention training, decentering, and cognitive strategies, and mind–body therapy [46].
Cognitive impairment	Six items from EORTC’s item library [24]. Cut-off ≤75: cognitive impairment	Recommended interventions include psychoeducation, energy management, compensatory strategies (planners, reminders), exercise, limiting alcohol intake, meditation, and cognitive training activities [47,48].

**Table 2 cancers-17-01127-t002:** Mean differences for MYCaW scores and symptom-specific PROMs’ scores.

MYCaW										
	1 Month Post-Intervention					6 Months Post-Intervention				
	Pre-Intervention Mean (95% Cl)	1 Month FU Mean (95% Cl)	Mean Difference *	*p*-Value	Hedges’s g #	Pre-Intervention Mean (95% Cl)	6 Month FU Mean (95% Cl)	Mean Difference **	*p*-Value	Hedges’s g #
Primary LE	4.70 (4.22–5.18)	3.20 (2.39–4.01)	**1.50 (0.66–2.34)**	**0.003**	1.54	4.50 (3.93–5.07)	2.67 (1.23–4.10)	**1.83 (0.44–3.23)**	**0.020**	1.69
Secondary LE	4.20 (3.90–4.50)	2.70 (1.87–3.53)	**1.50 (0.89–2.11)**	**<0.001**	1.65	4.33 (3.79–4.88)	2.83 (1.44–4.23)	**1.50 (0.40–2.60)**	**0.017**	1.63
General wellbeing	3.40 (2.56–4.24)	2.20 (1.54–2.86)	**1.20 (0.46–1.94)**	**0.005**	1.09	3.67 (2.23–5.10)	2.33 (1.25–3.42)	**1.33 (0.06–2.60)**	**0.042**	1.02
**Symptom-specific PROMs** ^†^										
	**1** Month Post-Intervention					**6** Months Post-Intervention				
	**Pre-intervention** **Mean (95% Cl)**	**1 month FU** **Mean (95% Cl)**	**Mean** **Difference ***	***p*-value**	**Hedges’s g #**	**Pre-intervention** **Mean (95% Cl)**	**6 month FU** **Mean (95% Cl)**	**Mean Difference ****	***p*-value**	**Hedges’s g #**
Anxiety ^§^	5.10 (1.95–8.25)	2.50 (1.32–3.68)	**2.60 (0.03–5.17)**	**0.048**	0.75	5.00 (−0.47–10.47)	3.50 (0.63–6.37)	1.50 (−2.84–5.84)	0.415	0.33
Depression ^§^	6.80 (3.86–9.74)	4.20 (2.36–6.04)	2.60 (−0.23–5.43)	0.067	0.73	8.00 (2.99–13.01)	5.33 (2.32–8.35)	2.67 (−2.15–7.48)	0.214	0.62
FCR^§^	15.90 (10.56–21.24)	13.70 (9.72–17.68)	2.20 (−1.43–5.83)	0.203	0.32	14.83 (8.63–21.03)	13.83 (5.09–22.57)	−1.00 (−8.06–6.06)	0.731	0.13
Insomnia ^§^	9.40 (6.14–12.66)	5.30 (2.32–8.28)	**4.10 (0.40–7.80)**	**0.033**	0.90	10.00 (4.77–15.22)	7.83 (3.36–12.30)	2.17 (−4.42–8.75)	0.436	0.43
Fatigue ^‡^	35.20 (28.77–41.63)	36.30 (29.27–43.33)	−1.1 (−5.98–3.78)	0.622	0.10	33.00 (18.52–47.48)	31.60 (14.65–48.55)	1.40 (−3.61–6.41)	0.481	0.10
Pain ^§^	10.50 (1.57–19.43)	8.40 (−0.25–17.05)	2.10 (−3.87–8.07)	0.447	0.16	5.67 (−3.92–15.25)	4.83 (−4.13–13.80)	−0.83 (−2.98–1.31)	0.363	0.09
Cognitive impairment ^‡^	47.22 (28.65–65.79)	61.11 (46.97–75.26)	−13.89 (−29.48–1.70)	0.075	0.58	51.11 (22.50–79.72)	56.67 (21.02–92.31)	−5.56 (−20.98–9.87)	0.373	0.20

Abbreviations: Cl = confidence interval; LE = late effect; FCR = fear of cancer recurrence; FU = follow-up; PROMS = patient reported outcome measures. Scheme 05. level marked with bold. Mean values and mean difference are presented with 95% Cl. * Mean difference between pre-intervention mean and 1 month FU mean is calculated as paired observations for the number of patients completing both these questionnaires at these timepoints (*n* = 10). ** Mean difference between pre-intervention mean and 6 month FU mean is calculated as paired observations for the number of patients completing both these questionnaires at these timepoints (*n* = 6). ^†^ Symptom-specific PROMs. Anxiety: Generalized Anxiety Disorder-7 (GAD-7). Depression: Patient Health Questionnaire-p (PHQ-9). FCR: FCR Inventory—short form (FCRI-SF). Insomnia: The Insomnia severity index (ISI). Fatigue: Functional Assessment of Chronic Illness Therapy—Fatigue’ (FACIT-F). Pain: The Rectal cancer pain score. Cognitive impairment: Six items from EORTC’s item library. ^‡^ Higher scores indicate less symptomatology; ^§^ Higher scores indicate greater symptomatology; # Effect size: Hedges’s g. A positive value indicates an effect size in the hypothesized direction. Conventions: small (0.2), medium (0.5), large (0.8).

## Data Availability

Data available upon reasonable request.

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
