# Peer review of "Psychological Intervention for Patients with Biopsychosocial Late Effects Following Surgery for Colorectal Cancer with Peritoneal Metastases—A Feasibility Study"

_cancers, 2025, doi:10.3390/cancers17071127_

Round 1
Reviewer 1 Report
Comments and Suggestions for Authors
Thanks to the authors of the article for paying attention to end-stage patients and managing their treatment and psychological rehabilitation.
Comments:
Title: Acceptable.
Introduction: Explain more fully about the psychological symptoms of cancer patients and the problems of the families of these patients.
Methods:
Include and exclude criteria should be stated more clearly.
The number of samples is not enough for final conclusions.
If there is a control group, the results will be analyzed more precisely.
Transparent psychological interview(individual or group) and executor skills.
How much did the amount of loss in the study according to the type of patients(cancer) .
Conclusion: the number of samples is not sufficient for appropriate conclusion.
Author Response
General comment:
Thanks to the authors of the article for paying attention to end-stage patients and managing their treatment and psychological rehabilitation.
Our response:
We appreciate the reviewer´s comment.
Specific comments
Comment 1.1.: Introduction: Explain more fully about the psychological symptoms of cancer patients and the problems of the families of these patients
Our response:
The psychological symptoms of cancer have now been explained in more detail. See page 4, lines 70-75. We agree that the impact these late effects may have on families is an important aspect of survivorship. However, this aspect has not been our focus in the present paper. The aim of our study was to assess the feasibility of the intervention offered. The following text has now been included:
For colorectal cancer (CRC) patients (stage I-III), the literature shows that many of these patients may suffer from poor sleep quality and that this is associated with an increased incidence of anxiety and depression[3]. FCR is another well-known LE in CRC survivors with a review from 2021 reporting that up to 16% of patients experience a high level of FCR[5]. Cognitive impairment, e.g., difficulties with concentration and memory, is another LE especially affecting those returning to the labor market [8]. In a recent paper, we have investigated these and other LEs in CRC stage IV patients [10].
Comment 1.2.: Methods: Include and exclude criteria should be stated more clearly.
Our response:
We thank the reviewer for pointing this out. Inclusion and exclusion criteria were the same as in our previous study which is referred to in the beginning of the introduction (reference 10). However, this is not sufficient, and we have now included the following on page 6 lines 124-126:
The inclusion criteria were age ≥ 18 years, the ability to understand written Danish and undergoing curatively intended surgery for CRC stage IV (peritoneal metastases). The exclusion criteria were patients undergoing surgery for cancers other than CRC stage IV.
Comment 1.3.: The number of samples is not enough for final conclusions.
Our response:
We agree and thank the reviewer for bringing this to our attention. The primary aim was to test the feasibility of the intervention approach offered. Although the number of patients is small, the results indicate that for these patients the intervention approach was shown to be feasible and acceptable. Being a feasibility study with a limited sample size, we cannot conclude that patients improved and benefitted from the intervention. Hence, we have now rewritten the conclusion and have omitted the sentence “appeared to benefit from them”. See page 16, lines 372-373:
Patients successfully engaged in the MDT conferences, initiated, and completed the personalized interventions offered.
Comment 1.4.: If there is a control group, the results will be analyzed more precisely.
Our response:
We, of course, agree that control groups are important when assessing the efficacy of interventions. In the present preliminary feasibility study, our aim was to evaluate the practical feasibility and acceptability of the MDT examination and personalized intervention approach, and we did not aim to test and quantify efficacy. In fact, in a small feasibility study, any effects compared to a control group would lack sufficient precision and be insufficiently powered to avoid Type-2 error. The next step will, of course, be to test the efficacy of the approach in a larger, sufficiently powered study.
Comment 1.5.: How much did the amount of loss in the study according to the type of patients(cancer).
Our response:
All the included patients only had surgery for one type of cancer – colorectal cancer with peritoneal metastases (stage IV colorectal cancer).
Comment 1.6.: Conclusion: the number of samples is not sufficient for an appropriate conclusion
Our response:
We agree with the reviewer. See our responses to comment 1.3. and 1.4.

Reviewer 2 Report
Comments and Suggestions for Authors
The authors examined the feasibility and outcomes of a treatment strategy for LE in this study. The article describes patients from Denmark who were examined for biopsychosocial LEs (anxiety, depression, fear of cancer recurrence, insomnia, cognitive impairment, pain, fatigue).
The introduction comprehensively introduces the subject of the article. The objectives of the study were correctly stated. The materials and methods and the results constitute a logical whole and correlate with the stated objectives. The results in the form of figures and tables correctly present the obtained results. The discussion presents the most important obtained results and the research of others on this subject.
The conclusions correlate with the obtained results. As indicated by the authors of the article, screening and treatment of biopsychosocial LE should be part of regular, comprehensive follow-up care for patients with cancer. The topic taken up by the authors is rarely addressed. Considering that colorectal cancers are a significant problem in society, the article is worth publishing. This topic may increase public awareness of the implementation of interventions in patients with colorectal cancer.
I have no major comments on the article. The study can be considered a pilot study.
A minor comment:
A small sample size of people was included in the intervention, only 13 people took part in the MDT and were offered a personalized intervention. Did the authors calculate the sample size?
It is worth adding a paragraph about future implications to the discussion.
Author Response
General comments:
The authors examined the feasibility and outcomes of a treatment strategy for LE in this study. The article describes patients from Denmark who were examined for biopsychosocial LEs (anxiety, depression, fear of cancer recurrence, insomnia, cognitive impairment, pain, fatigue).
The introduction comprehensively introduces the subject of the article. The objectives of the study were correctly stated. The materials and methods and the results constitute a logical whole and correlate with the stated objectives. The results in the form of figures and tables correctly present the obtained results. The discussion presents the most important obtained results and the research of others on this subject. The conclusions correlate with the obtained results. As indicated by the authors of the article, screening and treatment of biopsychosocial LE should be part of regular, comprehensive follow-up care for patients with cancer. The topic taken up by the authors is rarely addressed. Considering that colorectal cancers are a significant problem in society, the article is worth publishing. This topic may increase public awareness of the implementation of interventions in patients with colorectal cancer. I have no major comments on the article. The study can be considered a pilot study.
Our response:
We appreciate the reviewer´s positive remarks.
Specific comments
Comment 2.1.: A small sample size of people was included in the intervention, only 13 people took part in the MDT and were offered a personalized intervention. Did the authors calculate the sample size?
Our response:
Thank you for the question. The aim of the present study was to evaluate if the MDT conference examination and subsequent personalized intervention approach was feasible. We did not aim to test the efficacy of the intervention at this stage, and therefore did not conduct a statistical power analysis with the aim of estimating a sample size. The next step will, of course, be to test the efficacy of the approach in a larger, sufficiently powered study.
Comment 2.2.: It is worth adding a paragraph about future implications to the discussion.
Our response:
We thank the reviewer for the suggestion. We have now added a paragraph on future perspectives to the manuscript on page 16, lines 351-357:
The intervention in the present study (conducting an online MDT conference and online personalized psychological intervention addressing the most debilitating LEs) is part of an ongoing Danish national study for CRC stage I-II. The study is called RESPONSE: Colorectal Cancer Survivors’ Follow-up Care – Now Digital and Need-based. The overall aim of this study is to investigate whether a change to the current follow-up program following treatment for CRC stage I-II could improve health-related QoL without compromising overall survival, recurrence-free survival or increasing costs. As part of the change in the current follow-up program, the approach described in the present feasibility study will be further evaluated. This study has been preregistered with ClinicalTrials.gov (trial #: NCT06614647).
